# Extrapyramidal side effects of antipsychotics are linked to their association kinetics at dopamine $D_2$ receptors

David A. Sykes[1], Holly Moore[2,3], Lisa Stott[1], Nicholas Holliday[1], Jonathan A. Javitch[2,4,5], J. Robert Lane[6] & Steven J. Charlton [1]

Atypical antipsychotic drugs (APDs) have been hypothesized to show reduced extra-pyramidal side effects (EPS) due to their rapid dissociation from the dopamine $D_2$ receptor. However, support for this hypothesis is limited to a relatively small number of observations made across several decades and under different experimental conditions. Here we show that association rates, but not dissociation rates, correlate with EPS. We measured the kinetic binding properties of a series of typical and atypical APDs in a novel time-resolved fluorescence resonance energy transfer assay, and correlated these properties with their EPS and prolactin-elevating liabilities at therapeutic doses. EPS are robustly predicted by a rebinding model that considers the microenvironment of postsynaptic $D_2$ receptors and integrates association and dissociation rates to calculate the net rate of reversal of receptor blockade. Thus, optimizing binding kinetics at the $D_2$ receptor may result in APDs with improved therapeutic profile.

[1] School of Life Sciences, Queen's Medical Centre, University of Nottingham, Nottingham NG7 2UH, UK. [2] Department of Psychiatry, Columbia University, New York, NY 10032, USA. [3] Integrative Neuroscience, New York State Psychiatric Institute, New York, NY 10032, USA. [4] Department of Pharmacology, Columbia University, New York, NY 10032, USA. [5] Molecular Therapeutics, New York State Psychiatric Institute, New York, NY 10032, USA. [6] Drug Discovery Biology, Monash Institute of Pharmaceutical Sciences, Monash University, 381 Royal Parade, Parkville, 3052 VIC, Australia. Correspondence and requests for materials should be addressed to S.J.C. (email: steven.charlton@nottingham.ac.uk)

Imbalances in dopamine signaling are believed to play an integral part in the symptoms of schizophrenia. The efficacy of all currently marketed antipsychotic drugs (APDs) is thought to be mediated by attenuation of dopamine transmission through their actions as antagonists or low efficacy partial agonists at the dopamine $D_2$ receptor ($D_2R$)[1, 2]. However, the therapeutic window, i.e., the margin between the therapeutic dose and the dose that produces adverse side effects, varies considerably across these drugs[1–6].

Extrapyramidal motor symptoms (e.g., acute dystonia and parkinsonian symptoms such as bradykinesia and tremor) and excess prolactin release are major adverse side effects of APDs mediated by blockade of $D_2R$ signaling in the nigrostriatal dopamine system and the tuberoinfundibular pathway, respectively[1–6]. Many "typical" or first-generation antipsychotics (FGAs) exhibit a relatively narrow therapeutic window with respect to these "on-target" side effects. The term "atypical" was first applied to clozapine, an efficacious APD with markedly lower "on-target" side effects when compared to FGAs, but that carries the risk for agranulocytosis, a potentially life-threatening off-target toxicity. From the study of clozapine and FGAs emerged second-generation antipsychotics (SGAs) designed to exhibit wider therapeutic windows[3]. However, certain SGAs first introduced as atypical have in subsequent studies been shown to have therapeutic indices more consistent with typical APDs with the converse also true for FGA/typical APDs, such as melperone[7]. What is apparent is that despite more than 50 years of pharmacological research into APDs, on-target side effects remain a significant problem, often resulting in poor drug compliance. Thus, understanding their cause is a critical step toward the design of better therapeutics[2, 3, 8].

Several different pharmacological theories have been proposed to account for the atypicality of SGAs. One proposed mechanism is antagonism of the 5-$HT_{2A}$ receptor, which is thought to "balance" striatal dopamine signaling and thus reduce extrapyramidal side effects (EPS)[9–11]. However, the observation that the SGA amisulpride, which is considerably more $D_2$ selective over 5-$HT_{2A}$ yet still exhibits reduced EPS, suggests that this theory cannot account for all examples of atypicality[12, 13].

Another enduring theory of atypicality is based on the dissociation kinetics of APDs from the $D_2R$. This concept originated from the observation that some atypical APDs have lower affinity for the $D_2R$ than typical APDs[14–16], which was later demonstrated to be due to a faster dissociation rate[17–19]. This led Seeman and Kapur to propose the "fast off hypothesis," whereby rapid dissociation from the $D_2R$ contributes to the reduced side effect profile of atypical APDs[13]. Key to this hypothesis is the rapid and transient nature of synaptic dopamine signaling. Rapid dissociation of an antagonist will allow a greater fraction of $D_2Rs$ to be bound by the transiently high local concentrations of released dopamine, therefore out-competing the antagonist in a surmountable fashion. In contrast, an antagonist with a slow dissociation rate is unlikely to dissociate from the $D_2R$ in the short time frame between dopamine release and re-uptake, blocking the receptor regardless of the local concentration of dopamine that is achieved, i.e., making the antagonism effectively insurmountable[20].

The link between dissociation rate and "atypicality" has been questioned, however, based in part on the fact that the atypical APD olanzapine has relatively high affinity for the $D_2R$ and should, in theory, dissociate as slowly from the $D_2R$ as the typical APD haloperidol[3]. This inference is based on the widely held assumption that APDs exhibit similar association rates ($k_{on}$) for the $D_2R$ and therefore that affinity is essentially driven by differences in dissociation rate. Although association rates have widely been assumed to be diffusion limited, we recently

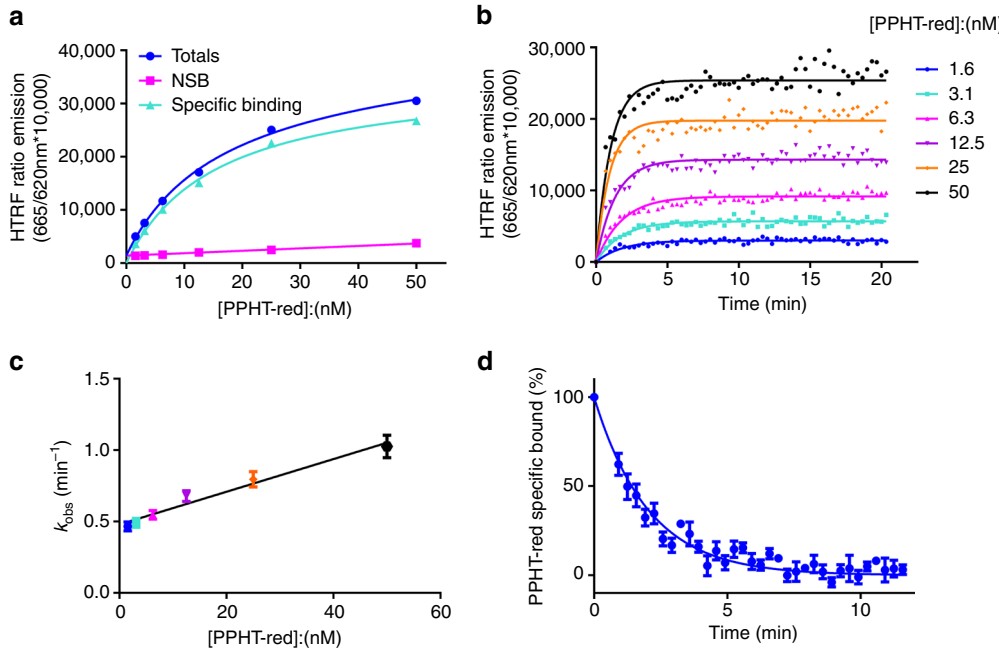

**Fig. 1** Determination of PPHT-red equilibrium and kinetic binding parameters. **a** Saturation analysis showing the binding of PPHT-red to the human dopamine $D_2R$. CHO–$D_2R$ cell membranes (2 μg per well) were incubated for 120 min with gentle agitation with increasing concentrations of PPHT-red. Data are presented in singlet form from a representative of 13 experiments. **b** Observed association of PPHT-red binding to the human dopamine $D_2R$. Data are presented in singlet form from a representative of 13 experiments. **c** Plot of PPHT-red concentration vs. $k_{obs}$. Binding followed a simple law of mass action model, $k_{obs}$ increasing in a linear manner with fluorescent ligand concentration. Data are presented as mean ± s.e.m. from a total of 13 experiments. **d** PPHT-red dissociation following addition of haloperidol (10 μM). Dissociation data are presented in mean ± s.e.m. from four experiments performed in singlet. All binding reactions were performed in the presence of GppNHp (100 μM) with nonspecific-binding levels determined by inclusion of haloperidol (10 μM)

**Table 1 Kinetic binding parameters of unlabeled dopamine $D_2$ antagonists for human $D_{2L}$ receptors and their historical classification as atypical or typical and those characterized as typical/atypical**

|  | $k_{on}$ (min$^{-1}$) | $k_{on}$ (M$^{-1}$ min$^{-1}$) | $t_{1/2}$ (min) | $pK_d$ | $pK_i$ |
|---|---|---|---|---|---|
| *SGA/Atypical* |  |  |  |  |  |
| Paliperidone | 0.44 ± 0.04 | 1.80 ± 0.29 × 10$^8$ | 1.58 | 8.60 ± 0.07 | 8.54 ± 0.07 |
| Remoxipride | 1.90 ± 0.55 | 1.16 ± 0.37 × 10$^7$ | 0.36 | 6.79 ± 0.04 | 6.68 ± 0.07 |
| Clozapine | 1.67 ± 0.25 | 8.23 ± 1.42 × 10$^7$ | 0.41 | 7.69 ± 0.02 | 7.60 ± 0.02 |
| Ziprasidone | 1.07 ± 0.52 | 1.61 ± 0.50 × 10$^9$ | 0.65 | 9.19 ± 0.18 | 9.16 ± 0.19 |
| Risperidone | 0.43 ± 0.05 | 4.38 ± 0.52 × 10$^8$ | 1.61 | 9.05 ± 0.01 | 8.95 ± 0.05 |
| Sertindole | 0.59 ± 0.01 | 4.91 ± 0.82 × 10$^8$ | 1.17 | 8.92 ± 0.07 | 8.89 ± 0.03 |
| Quetiapine | 1.01 ± 0.33 | 6.57 ± 0.85 × 10$^6$ | 0.69 | 6.82 ± 0.02 | 6.75 ± 0.07 |
| Olanzapine | 1.12 ± 0.12 | 1.79 ± 0.44 × 10$^8$ | 0.62 | 8.17 ± 0.09 | 8.08 ± 0.11 |
| Asenapine | 0.93 ± 0.06 | 2.17 ± 0.65 × 10$^9$ | 0.75 | 9.29 ± 0.16 | 9.29 ± 0.12 |
| Amisulpride | 0.83 ± 0.05 | 3.44 ± 0.50 × 10$^8$ | 0.83 | 8.61 ± 0.06 | 8.49 ± 0.07 |
| *FGA/Typical* |  |  |  |  |  |
| (+)Butaclamol | 0.026 ± 0.004 | 6.82 ± 2.44 × 10$^8$ | 26.65 | 10.37 ± 0.12 | 10.32 ± 0.13 |
| Flupenthixol | 0.072 ± 0.010 | 3.50 ± 0.72 × 10$^8$ | 9.63 | 9.67 ± 0.07 | 9.68 ± 0.10 |
| Haloperidol | 0.65 ± 0.07 | 2.13 ± 0.52 × 10$^9$ | 1.07 | 9.49 ± 0.08 | 9.48 ± 0.08 |
| Fluphenazine | 0.040 ± 0.004 | 1.13 ± 0.01 × 10$^9$ | 17.33 | 10.46 ± 0.06 | 10.29 ± 0.04 |
| Chlorpromazine | 2.20 ± 0.44 | 3.76 ± 0.70 × 10$^9$ | 0.32 | 9.24 ± 0.04 | 9.01 ± 0.11 |
| Perphenazine | 0.23 ± 0.02 | 1.29 ± 0.17 × 10$^9$ | 3.01 | 9.73 ± 0.06 | 9.53 ± 0.04 |
| Trifluoperazine | 0.22 ± 0.01 | 1.10 ± 0.20 × 10$^9$ | 3.15 | 9.69 ± 0.10 | 9.48 ± 0.04 |
| Spiperone | 0.038 ± 0.006 | 2.55 ± 0.12 × 10$^9$ | 18.24 | 10.84 ± 0.07 | 10.54 ± 0.04 |
| Nemonapride | 0.018 ± 0.001 | 1.44 ± 0.04 × 10$^9$ | 38.50 | 10.91 ± 0.03 | 10.39 ± 0.04 |
| Droperidol | 0.38 ± 0.04 | 2.01 ± 0.38 × 10$^9$ | 1.82 | 9.71 ± 0.10 | 9.99 ± 0.10 |
| *Typical/atypical* |  |  |  |  |  |
| (−)Sulpiride | 2.23 ± 0.93 | 1.60 ± 0.67 × 10$^8$ | 0.31 | 7.87 ± 0.06 | 7.58 ± 0.10 |
| Thioridazine | 1.41 ± 0.25 | 2.37 ± 0.61 × 10$^9$ | 0.49 | 9.21 ± 0.03 | 8.93 ± 0.05 |
| Molindone | 1.69 ± 0.45 | 8.69 ± 2.6 × 10$^7$ | 0.41 | 7.69 ± 0.10 | 7.57 ± 0.12 |
| Loxapine | 2.14 ± 0.33 | 4.04 ± 1.04 × 10$^8$ | 0.32 | 8.25 ± 0.05 | 8.18 ± 0.05 |
| Raclopride | 0.53 ± 0.14 | 6.69 ± 2.04 × 10$^8$ | 1.31 | 9.08 ± 0.04 | 8.91 ± 0.08 |
| Melperone | 1.48 ± 0.18 | 1.99 ± 0.41 × 10$^7$ | 0.47 | 7.11 ± 0.04 | 7.05 ± 0.03 |
| Zotepine | 1.41 ± 0.34 | 9.15 ± 1.63 × 10$^8$ | 0.49 | 8.76 ± 0.17 | 8.69 ± 0.11 |
| *Peripheral acting* |  |  |  |  |  |
| Domperidone | 0.14 ± 0.02 | 9.73 ± 1.66 × 10$^8$ | 4.95 | 9.83 ± 0.07 | 9.81 ± 0.09 |

Data are mean ± s.e.m. from four experiments performed in singlet. FGA/typical and SGA/atypical classification is based on reference sources[2, 3, 30]. A number of APDs have been classified as both typical and atypical APDs in separate studies. To recognize this, we have placed these drugs in a third group "typical/atypical" that includes sulpiride[60–62], melperone[7, 62, 63], loxapine[58, 59, 64], molindone[65–67], zotapine[7, 61], raclopride[68, 69], and thioridazine[62, 66, 70]. Domperidone is not an APD and is used to block $D_2$ receptors in the periphery

found that the association rates differ by several orders of magnitude across a range of structurally diverse $D_2R$ agonists[21], demonstrating that the mechanisms that determine association rate can vary greatly with ligand structure.

The majority of drug-receptor-binding models assume free diffusion of analytes such that the dynamics of the system are reaction-limited. In certain tissue microenvironments, however, this assumption may not be valid, due, in part, to limitations on free diffusion arising from physical barriers. For example, the small aqueous compartment within a dopamine synapse (estimated to be 0.09–0.4 μm$^3$)[22] is unlikely to mix well with the bulk aqueous phase surrounding the synapse under the temporal and spatial scales over which neurotransmission operates. This may have important implications with regard to the blockade of dopamine synaptic signals and the ability of APDs to rebind free receptors. Rebinding in this case describes the process whereby a reversibly bound ligand dissociates from a receptor into the local aqueous environment but then rebinds the same or a nearby receptor before it is able to diffuse from the synaptic cleft[23], effectively maintaining a higher concentration of the drug near the receptor. Under these conditions, the degree to which an individual drug rebinds is determined by receptor density, the association rate constant, and anatomical and physicochemical factors affecting the diffusion of the ligand away from the receptor[24].

Currently available equilibrium and kinetic data on the binding of APDs to the $D_2R$ were derived over the past several decades

using an assortment of different methods. The most common method has been to use radiolabeled compounds[18, 19, 25], although not all APDs are available as radioligands. Alternatively, competition association assays formulated with a single radioligand/tracer can enable the kinetics of unlabeled ligands to be calculated[26, 27]. We have recently developed such an assay utilizing time-resolved fluorescence resonance energy transfer (TR-FRET) to measure the binding kinetics of unlabeled $D_2R$ agonists[21, 28]. In the present study, we use this method to determine the kinetics of an extensive series of APDs under physiological temperature and sodium ion concentration, and in doing so explore the kinetic basis for on-target side effects. We find that association rates, but not dissociation rates, correlate with EPS. EPS were predicted by a rebinding model that integrates association and dissociation rates within the microenvironment of postsynaptic $D_2Rs$ to calculate the net rate of reversal of receptor blockade. In contrast, prolactin elevation was directly correlated with APD off-rate from $D_2R$. Thus, optimizing binding kinetics at the $D_2R$ may result in APDs with improved therapeutic profile.

### Results

**Characterization of PPHT-red binding**. Specific binding of the agonist PPHT-red to human $D_{2L}$ receptor (h$D_{2L}$R) expressed in CHO membranes was saturable and best described by the interaction of the fluorescent ligand with a single population of

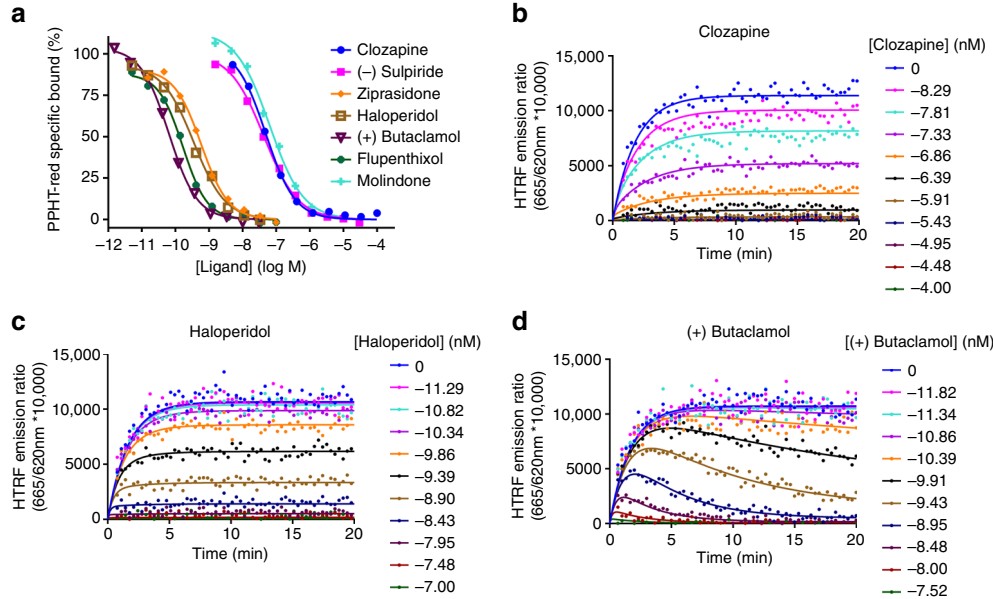

**Fig. 2** Equilibrium and competition association binding. **a** Competition between PPHT-red (12.5 nM) and increasing concentrations of representative atypical and typical APDs clozapine, (−)sulpride, ziprasidone, haloperidol, (+)butaclamol, fluphenthixol, and molindone at the human dopamine $D_2R$. PPHT-red competition association curves in the presence of **b** clozapine, **c** haloperidol, and **d** (+)butaclamol. All binding reactions were performed in the presence of GppNHp (100 μM) with nonspecific-binding levels determined by inclusion of haloperidol (10 μM). Kinetic and equilibrium data were fitted to the equations described in "Methods" section to calculate $K_i$, $K_d$, and $k_{on}$ and $k_{off}$ values for the unlabeled ligands; these are summarized in Table 1. Data are presented as singlet values from a representative of four. All data used in these plots are detailed in Table 1

binding sites (Fig. 1a). From these studies, the equilibrium dissociation constant ($K_d$) of PPHT-red was determined to be $16.3 \pm 0.9$ nM. The expression level of the $hD_{2L}R$ recombinantly expressed in CHO cells was assessed, using [$^3$H]-spiperone saturation binding and determined to be $1.13 \pm 0.11$ pmol mg$^{-1}$ protein.

The binding kinetics of PPHT-red were characterized by monitoring the observed association rates at six different ligand concentrations (Fig. 1b). The observed rate of association was related to PPHT-red concentration in a linear fashion (Fig. 1c). Kinetic rate parameters for PPHT-red were calculated by globally fitting the association time courses, resulting in a $k_{on}$ of $2.3 \pm 0.14 \times 10^7$ M$^{-1}$ min$^{-1}$ and a $k_{off}$ of $0.33 \pm 0.01$ min$^{-1}$. The resulting $K_d$ ($k_{off}/k_{on}$) of $15.4 \pm 0.11$ nM was comparable to that obtained from the equilibrium studies. Ligand dissociation estimated directly through addition of an excess of haloperidol revealed a $k_{off}$ value of $0.52 \pm 0.04$ min$^{-1}$, which was in good agreement with the value estimated from the global association time course described above (Fig. 1d).

The binding affinity of the various ligands for the $hD_{2L}R$ was measured at equilibrium at 37 °C in a buffer containing 5′-guanylyl imidodiphosphate (GppNHp) (0.1 mM) to ensure that antagonist binding only occurred to the G protein-uncoupled form of the receptor. Binding affinities ($K_i$ values) for the APDs studied are summarized in Table 1, and representative competition curves are presented in Fig. 2a. In this table we have separated APDs into those described in literature as typical APDs, those described as atypical and, in a third group, those that have been described as both typical and atypical.

Representative kinetic competition curves for selected $D_2R$ ligands are shown in Fig. 2b–d. Progression curves for PPHT-red alone and in the presence of competitor were globally fitted to Eq. 3 enabling the calculation of both $k_{on}$ (k3) and $k_{off}$ (k4) for each of the ligands, as reported in Table 1.

There was a very wide range in dissociation rates for the different ligands, with $t_{1/2}$ values between 0.32 min for

chlorpromazine to 38.5 min for nemonapride. To validate the rate constants, the kinetically derived dissociation constant ($K_d$) values ($k_{off}/k_{on}$) were compared with the dissociation constant ($K_i$) obtained from equilibrium competition binding experiments (see Supplementary Fig. 1). There was a very good correlation between these two values for all APDs tested (two-tailed Pearson's correlation $r^2 = 0.98$, $P < 0.0001$) indicating the kinetic parameters were accurate. Previous radioligand-binding studies have reported differences in dissociation rates of the order of 100-fold between the typical APD chlorpromazine and the atypical drugs, clozapine and quetiapine[18, 19, 29]; however, the present study did not corroborate these findings. The most plausible explanation for the differences observed between this and the original studies is the use of different temperatures to study the kinetics of these compounds coupled with the use of a subsaturating concentration of dopamine (100 nM) as the cold competing ligand, which may not be sufficient to fully prevent rebinding of the radioligand[18, 29, 30].

**Comparing kinetics and on-target side effects of APDs.** Historically, the link between APD $D_2R$ affinity and dissociation rate is based on the assumption that APDs exhibit approximately the same $k_{on}$ for $D_2Rs$[19]. However, we observed an increased range in $k_{on}$ values between the atypical APDs compared to a relatively small range in $k_{off}$ values. In contrast, for typical APDs there was a much narrower variation in the value of $k_{on}$ and differences in affinity were driven instead by changes in $k_{off}$. Notably, however, the typical APD chlorpromazine exhibits $k_{off}$ values similar to, or indeed faster than atypical APDs (Table 1).

We have correlated our kinetic binding data with clinical findings taken from a recent meta-analysis of multiple-treatments studies comparing side effect profiles across a diverse group of APDs[31]. Both prolactin elevation and EPS are "on-target" adverse side effects related to $D_2R$ blockade, whereas adverse effects, such as QT interval and sedation, are generally considered "off-target" effects.

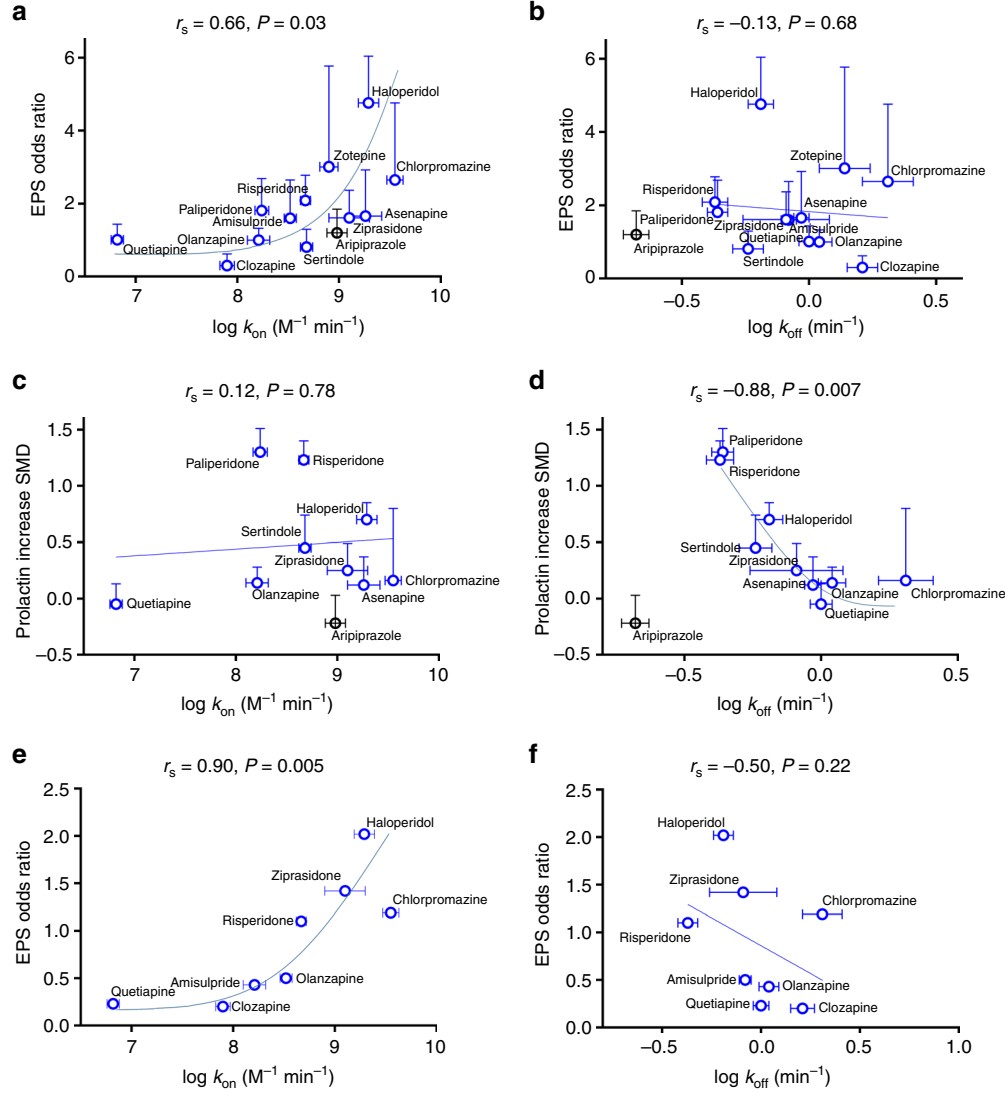

**Fig. 3** Correlating clinical data on APD "on-target" effects with kinetically derived parameters. Correlation plots showing the relationship between **a** log $k_{on}$ and EPS odds ratio and **b** log $k_{off}$ and EPS odds ratio and **c** log $k_{on}$ and prolactin increase and **d** log $k_{off}$ and prolactin increase. All kinetic data used in these plots are detailed in Table 1 and clinical data are taken from Leucht et al.[31]. Kinetic data for aripiprazole were taken from Klein-Herenbrink et al.[21]. Aripiprazole was not included in the correlation analysis as it is a dopamine $D_2R$ partial agonist. Correlation plot showing the relationship between **e** log $k_{on}$ and EPS odds ratio and **f** log $k_{off}$ and EPS odds ratio, clinical data taken from first-episode patient[50–57]. Kinetic data are presented as mean ± s.e.m. from four experiments and clinical data as standardized mean difference (*SMD*) for prolactin increase and odds ratio for EPS with associated credible intervals where indicated. The relationship between two variables was assessed using a two-tailed Spearman's rank correlation allowing the calculation of the correlation coefficient, $r_s$. A P value of 0.05 was used as the cutoff for statistical significance and relationships depicted as trend lines

In contrast with the "rapid dissociation hypothesis"[13], the kinetic $k_{on}$, but not $k_{off}$, was positively correlated with the incidence of EPS (Fig. 3a, b, Spearman's $r_s = 0.68$, $P < 0.05$ and $r_s = -0.13$, $P = 0.68$, respectively). On the other hand, prolactin increases were correlated with the kinetic $k_{off}$ but not the $k_{on}$ (see Fig. 3d, c, Spearman's $r_s = -0.82$, $P < 0.05$ and $r_s = 0.12$, $P = 0.78$, respectively). An obvious outlier is the atypical APD aripiprazole, which displays an extremely slow $k_{off}$[21] but displays very little propensity to cause prolactin release. It should be noted that the mechanism of action of aripiprazole is different from other APDs in that it is a low efficacy partial agonist[32, 33]. As expected, QT prolongation, an off-target side effect, correlated neither with $k_{on}$ nor $k_{off}$ (Spearman's $r_s = 0.09$, $P = 0.81$ and $r_s = 0.30$, $P = 0.41$, respectively; see Supplementary Fig. 2a, b).

The majority of studies covered in the Leucht et al.[31] meta-analysis included as subjects chronically ill patients with a history of APD exposure, which has been previously shown to change

$D_2R$ availability[34]. To address this issue, we used data from available multiple treatment studies in first-episode patients to test correlations of $k_{on}$ and $k_{off}$ with EPS odds ratios (ORs) in patients with minimal or no APD exposure. For the APDs tested in these studies, we found that ORs of EPS were robustly predicted by $k_{on}$ (Spearman's $r_s = 0.90$, $P < 0.01$) but not significantly related to $k_{off}$ (Spearman's $r_s = -0.50$, $P = 0.22$, see Fig. 3e, f).

**Modeling rebinding at the $D_2R$.** Synapses are essentially minute gaps across which a neurotransmitter diffuses and as such can be considered receptor micro-compartments. While dopamine terminals rarely form classical synapses, they do form appositions with $D_2R$-expression domains of target neurons that likely impose diffusion constraints on drugs in these regions[35, 36].

Models of receptor rebinding in situations with limited diffusion allow the effect of $k_{on}$ and $k_{off}$ on the reversal of

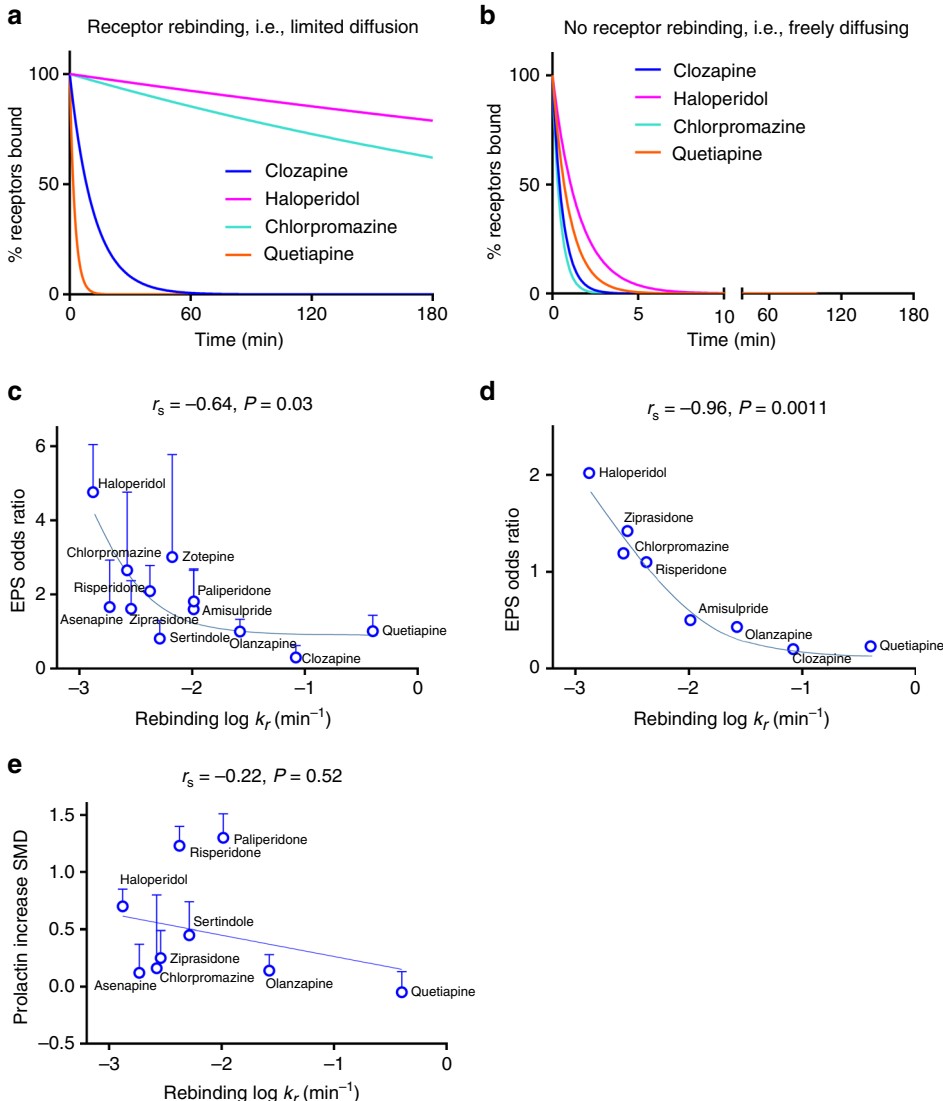

**Fig. 4** Modeling APD $D_2R$ rebinding and its consequences for clinical "on-target" toxic effects. Simulated dissociation rates of clinically relevant APs to human $D_2R$, **a** under conditions of limited diffusion based on the association ($k_{on}$) and dissociation ($k_{off}$) rates determined in competition kinetic binding experiments, **b** under condition of free diffusion based on measured off rates ($k_{off}$) determined in competition kinetic binding experiments. All kinetic parameters used to these plots are detailed in Table 1 and in the methods section associated with Eq. (4). For simulation purposes, the reversal rate $k_r$ was based on the model of an immunological synapse as detailed in the "Methods" section. Correlating clinical "on-target" effects with the kinetically derived overall reversal rate $k_r$. Correlation plot showing the relationship between **c** log $k_r$ and EPS odds ratio, taken from Leucht et al.[31] Correlation plot showing the relationship between **d** log $k_r$ and EPS odds ratio (relative to placebo or baseline conditions, averaged across studies), taken from studies of early psychosis patients[50-57]. Correlation plot showing the relationship between **e** log $k_r$ and prolactin increase, taken from Leucht et al.[31] All kinetic data used in these plots are detailed in Table 1. Kinetic data are presented as mean ± s.e.m. from four experiments and clinical data as standardized mean difference (SMD) for prolactin increase and odds ratio for EPS with associated credible intervals where indicated. The relationship between two variables was assessed using a two-tailed Spearman's rank correlation allowing the calculation of the correlation coefficient, $r_s$. A P value of 0.05 was used as the cutoff for statistical significance and relationships depicted as trend lines

antagonist receptor occupancy to be considered collectively to derive an overall reversal rate ($k_r$) that provides a measure of the local duration of antagonist effect[24]. The potential for rebinding of APDs was modeled according to two different scenarios; under conditions of limited diffusion (see Fig. 4a) such as those encountered at the level of a synapse; or with free diffusion (see Fig. 4b). In accordance with our current understanding of the rebinding process, $k_{on}$ was the dominant factor in determining the duration of target–receptor occupancy under conditions of limited diffusion. In contrast, $k_{on}$ had little effect on-target residency under conditions of free diffusion.

Consistent with our model, the correlation between $k_{on}$ and the incidence of EPS observed in the Leucht study was mirrored by the reversal rate $k_r$, suggesting that $k_{on}$ is important in dictating the reversal of $D_2R$ occupancy at the level of the synapse (Spearman's $r_s = -0.64$, $P < 0.05$, see Fig. 4c). Interestingly, this correlation was marginally stronger for first-episode patients receiving APDs (Spearman's $r_s = -0.95$, $P < 0.01$, see Fig. 4d). In contrast, $k_r$ was not significantly correlated with elevations of prolactin (which is in agreement with the lack of correlation with $k_{on}$) (Spearman's $r_s = -0.17$, $P = 0.68$, see Fig. 4e).

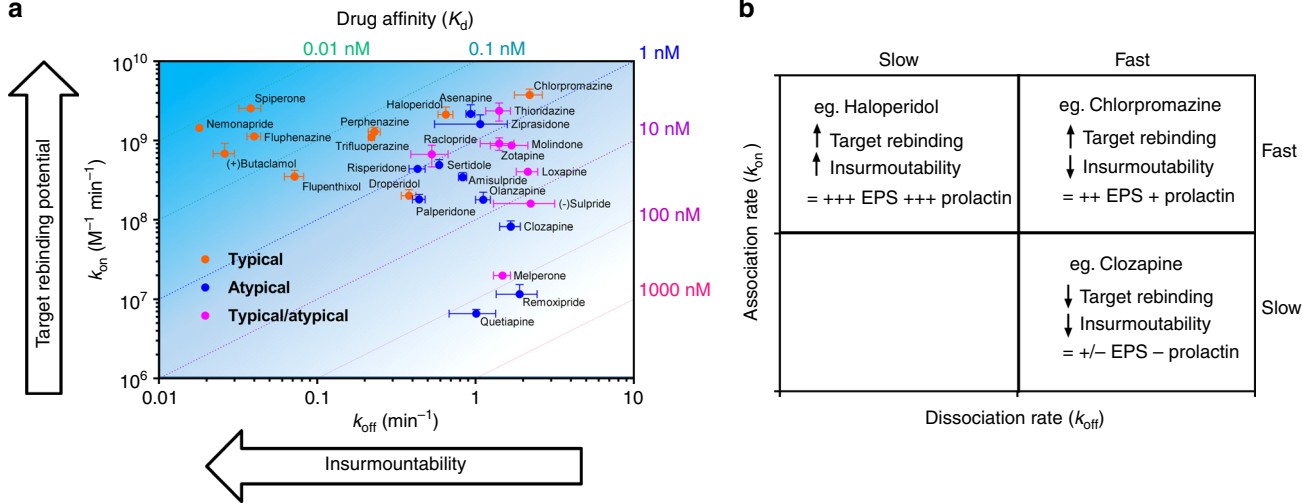

**Fig. 5** Summarizing the role of kinetics and rebinding in dictating the "on-target" AP toxicity. **a** APD $D_2R$ kinetic map showing SGA/atypical (*blue*), FGA/typical (*red*), and APDs described as both typical and atypical (*green*) plotted using their respective dissociation rate ($k_{off}$) and association ($k_{on}$) constants, with the combinations of $k_{off}$ and $k_{on}$ that result in identical affinity ($K_d$) values represented by *diagonal dotted lines*. The *arrows* on graph indicate the directions of increasing rebinding potential and insurmountability (due to hemi-equilibrium) dictated by $k_{on}$ and $k_{off}$, respectively, with the heat map representing the overall rate of binding reversal ($k_r$) from the $D_2R$. **b** Three types of APD are identified from this kinetic study and represented in the *box plot* along with their relative potential for "on-target" toxic effects indicated by the following; (−) no evidence, (+) some evidence, moderate (++) and (+++) strong evidence. Kinetic values are presented as mean ± s.e.m. from four experiments

## Discussion

The novel TR-FRET kinetic assay described herein has a significantly improved throughput relative to more traditional radioligand binding assays. This has enabled us to accurately quantify, for the first time, the kinetic rate constants of a large number of unlabeled dopamine $D_2$ antagonists under identical test conditions, allowing us to better investigate a role for kinetics in the side effect liability of clinically used APDs.

It has been widely assumed that association rates for APDs are diffusion limited and therefore comparable, meaning that the dissociation rate determines their affinity[37, 38]. Our TR-FRET data, however, revealed a surprisingly wide range of both association and dissociation rates across the ligands studied, demonstrating the importance of directly measuring rate constants. For example, the suggestion that the high-affinity atypical APDs olanzapine and risperidone should have $k_{off}$ values similar to haloperidol[3] is not supported by our kinetic data.

We correlated our new kinetic binding data with clinical data that quantified the level of extrapyramidal side effects and hyperprolactinemia associated with a diverse group of clinical APDs. The clinical data were taken from a recent and relatively comprehensive multiple-treatment meta-analyses of antipsychotic drug efficacies[31] and a summary of the primary literature on studies of drug-free patients. While meta-analytic methods assure the best control for the quality of the data across studies, it is important to note that because their focus was efficacy, EPS were often not a well-controlled outcome of the studies included in the meta-analysis. Indeed, the present models, like preceding hypotheses on EPS, are limited by the relative paucity of studies comparing antipsychotic drugs in which EPS are a primary outcome controlled for the minimum dose required to treat the psychosis or for adjunct treatments. To address this, we performed an additional exploratory analysis of studies in first episode or early psychosis, drug-free patients.

Consistent with the fast-dissociation hypothesis of APD atypicality, we found that hyperprolactinemia was correlated with the dissociation rate ($k_{off}$), with ligands that were the slowest to dissociate from the $D_2R$ displaying the greatest liability for prolactin elevation in patients. Surprisingly, however, we found it

was $k_{on}$ and not $k_{off}$ that was correlated with the incidence of EPS. Thus, drugs that bind more rapidly have greater liability of EPS, challenging the hypothesis that dissociation rate is the sole determinant of a compound's liability to produce this side effect[13]. To illustrate, the typical APD chlorpromazine has a $k_{off}$ value similar to that of clozapine, but has much greater EPS liability. The increased propensity for EPS of chlorpromazine relates instead to its rapid association rate ($k_{on}$). A recent study by Sahlholm and colleagues is consistent with this interpretation. These authors used $D_2R$-evoked potassium channel activation to estimate receptor kinetics[39, 40]. Interestingly, the off rates determined by this indirect measurement, which were broadly consistent with the values obtained in the current study, did not distinguish between the typical and atypical APDs.

To further explore this finding, we employed a more holistic model of receptor binding that integrates both the association and dissociation rates in a system mimicking the environment of the synapse. This model assumes that diffusion of the drug out of the synapse is reduced by the physical barriers created by the pre- and post-synaptic membranes, effectively creating a compartment separate from the bulk aqueous phase. The consequence of this is that freshly dissociated ligands tend to remain in close proximity with membrane surfaces for longer, increasing the probability of a second binding event to the same or nearby receptor[24]. By using our measured association and dissociation rates in this model, we have estimated the overall relative rate of reversal of receptor blockade, termed $k_r$[24].

As illustrated in Fig. 4, compounds with similar affinity and/or $k_{off}$ do not share the same potential to rebind receptors, a process which in the face of limited diffusion and subsaturating dopamine concentrations is governed largely by $k_{on}$. Remarkably when we calculated the receptor reversal rate $k_r$ for each APD using the measured kinetic parameters, we found that this parameter was significantly correlated with the incidence of EPS, (Spearman's $r_s = −0.64$, $P < 0.05$ see Fig. 4c). The active standardized mean difference (SMD) produced a similar correlation with $k_{on}$ and $k_r$, strongly suggesting that the process of rebinding and associated EPS liability may be a limiting factor for treatment effectiveness

(see Supplementary Figs. 2c and 3a), ultimately leading to discontinuation of therapy (see Supplementary Figs. 2d and 3b). Intriguingly, the correlation of $k_r$ with EPS was stronger for the first-episode patients (Spearman's $r_s = -0.95$, $P < 0.01$ see Fig. 4c), which may reflect the fact that in this case EPS liability is taken from well-controlled study outcomes in patients receiving minimal adjunctive treatments.

We believe that there is no real improvement in correlation with $k_r$ over $k_{on}$ because for the particular properties of our model synapse, $k_{on}$ is the governing rate parameter for reversal of binding. In other systems, this might not be the case. For example, changes in the dimensions or receptor density of the compartment might reduce the impact of rebinding, meaning $k_r$ would be largely driven by $k_{off}$. In these situations, it would be expected that $k_r$ would be better correlated to EPS than $k_{on}$. Thus, $k_r$ represents a useful parameter with which to predict the receptor occupancy of a drug with known kinetic binding parameters under different degrees of limited diffusion.

Intriguingly, radioligand binding studies have demonstrated that subsaturating concentrations of dopamine less readily displace more rapidly associating radioligands, such as chlorpromazine, compared to more slowly associating radioligands, such as clozapine, despite their off rates being almost identical[18]. These observations taken together with our model indicate that rebinding maintains APD at a higher concentration in the synaptic (or appositional) compartment, resulting in a more effective competition for released dopamine. This, combined with the close correlation between rebinding rates and EPS, leads us to speculate that there is a minimum level of stimulation of postsynaptic $D_2Rs$ that must be maintained in order to avoid EPS. The ability of dopamine transmission to remain above this threshold in the presence of an APD is determined in large part by the APD's rebinding rate. It is important, however, to acknowledge that the data presented in this study do not rule out alternative mechanisms that may contribute to the overall side effect profile of APDs, e.g., agonism at 5-HT$_{1A}$[41].

Interestingly, and in contrast to EPS, prolactin elevation was not correlated with $k_r$, reflecting the lack of correlation with $k_{on}$. This may reflect that dopamine and APDs diffuse into the pituitary through the hypothalamic–pituitary portal system as opposed to a synaptic apposition[42–44]. Since ligands diffuse more freely around $D_2Rs$ on pituitary lactotrophs, their behavior conforms to the laws of mass action and rebinding may be negligible. As a consequence, the rate of reversal of APD-receptor occupancy, and thus excess prolactin release, will depend solely on the dissociation rate constant of APDs through the phenomenon of insurmountable antagonism.

To summarize, we propose to expand the kinetic hypothesis for APD side effects by considering not only the dissociation rate (and therefore propensity to display insurmountable antagonism), but also their association rate and potential for receptor rebinding, leading to increased competition with dopamine at the synapse (see Fig. 5a). Based on this scheme, we propose the following three broad classes of compounds to explain how these different kinetic characteristics may influence on-target side effects in different tissues:

1. Fast on, slow off compounds, e.g., haloperidol. The fast on rate results in a high receptor rebinding potential at $D_2Rs$ apposed to dopamine release sites in the striatum and therefore high EPS. In contrast, in the pituitary, the slow dissociation rate results in insurmountable antagonism at $D_2Rs$ leading to increased prolactin release.

2. Fast on, fast off compounds, e.g., chlorpromazine. Again, the fast on rate leads to high rebinding potential in the striatum and high EPS, but fast off rates result in surmountable antagonism and thus reduced propensity for hyperprolactinaemia.

3. Slow on, fast off compounds, e.g., clozapine. Slow on rates result in lower rebinding potential in the striatum and low EPS, and fast off rates lead to surmountable antagonism and reduced hyperprolactinemia.

This classification, summarized in Fig. 5b, suggests that slow on/fast off kinetics is the optimal kinetic profile for APDs targeting $D_2Rs$. Notably, the APD with the slowest association rate is quetiapine, with a $k_{on}$ more than an order of magnitude slower than clozapine. Curiously, this compound has been found to be less efficacious than risperidone and olanzapine in treatment of chronic schizophrenia[45]. Quetiapine's removal from the comparisons between $k_{on}$ and active SMD resulted in a much-improved correlation (Spearman's $r_s = -0.85$, $P < 0.001$; see Supplementary Fig. 2c), suggesting it may not produce sufficient rebinding for a robust clinical effect. This problem is likely to be further exacerbated by its short plasma half-life, which will further reduce its receptor coverage over the dosing period[46].

Moreover, our model and the relative positions of compounds in Fig. 5a appear to help rationalize why certain compounds originally introduced as SGA/atypical APDs have typical profiles, e. g., zotapine, and conversely why drugs originally classified as FGA/typical APDs, such as melperone, are capable of displaying atypical behavior. The reality is that the side effects of APDs comprise multiple, pharmacologically separable effects, hence a more rational approach to classification is to consider a continuum based on the specific pharmacology of a given side effect rather than the current dichotomous system. This has lead us to speculate that there is likely to be a kinetic "sweet spot" where rebinding is sufficient for efficacy but not enough to cause EPS. Through optimization of these kinetic parameters, it may be possible to develop a new generation of safer drugs for a disease that still has high unmet medical need.

## Methods

**Materials**. Tag-lite labeling medium (LABMED), SNAP-Lumi4-Tb, and the PPHT ((±)-2-(n-phenethyl-n-propyl)amino-5-hydroxytetralin hydrochloride;1-Naphthalenol,5,6,7,8-tetrahydro-6-[(2-phenylethyl)propylamino]) derivative labeled with a red fluorescent probe (PPHT-red) was obtained from Cisbio Bioassays (Bagnols-sur-Cèze, France). Ninety-six-well polypropylene plates (Corning) were purchased from Fisher Scientific UK (Loughborough, UK) and 384-well optiplate plates were purchased from PerkinElmer (Beaconsfield, UK). GppNHp, risperidone, chlorpromazine hydrochloride, quetiapine hemifumarate, ziprasidone hydrochloride monohydrate, zotepine, sertindole, thioridazine hydrochloride, fluphenazine dihydrochloride, molindone hydrochloride, loxapine succinate, perphenazine, trifluoperazine dihydrochloride, spiperone, (−)-sulpiride, droperidol, and (+)-butaclamol used in competition assays were obtained from Sigma-Aldrich (Poole, UK). Olanzapine, nemonapride, remoxipride hydrochloride, flupentixol dihydrochloride, paliperidone, amisulpride, melperone hydrochloride, clozapine, raclopride, domperidone, asenapine maleate and haloperidol hydrochloride used for competition assays were obtained from Tocris Bioscience (Avonmouth, Bristol).

**Cell culture**. The host Chinese hamster ovary (CHO) K1 cell line was provided by Prof. J. Baker, University of Nottingham (ATCC #CCL-61). This was transfected with the cDNA encoding a SNAP-tagged human dopamine $D_{2L}$ receptor (Genbank ref.: NM_000795), and a stable dilution-cloned cell line (CHO–$D_{2L}$) was established by zeocin resistance encoded by the plasmid vector (pcDNA3.1zeo$^+$, Invitrogen, Paisley UK). Cells were maintained in Dulbecco's modified Eagle's medium: Ham F12 (DMEM:F12) containing 2 mM glutamine (Sigma-Aldrich, Poole, UK) and supplemented with 10% fetal calf serum (Life Technologies, Paisley UK).

**Terbium labeling of SNAP-tagged $D_{2L}$ cells**. Cell culture medium was removed from the t175 cm$^2$ flasks containing confluent adherent CHO–$D_{2L}$ cells. Twelve milliliter of Tag-lite labeling medium containing 100 nM of SNAP-Lumi4-Tb was added to the flask and incubated for 1 h at 37 °C under 5% $CO_2$. Cells were washed 2× in PBS (GIBCO Carlsbad, CA) to remove the excess of SNAP-Lumi4-Tb then detached using 5 ml of GIBCO enzyme-free Hank's-based cell dissociation buffer (GIBCO, Carlsbad, CA) and collected in a vial containing 5 ml of DMEM:F12 containing 2 mM glutamine (Sigma-Aldrich) and supplemented with 10% fetal calf serum. Cells were pelleted by centrifugation (5 min at 1500 rpm) and the pellets were frozen to −80 °C. To prepare membranes, homogenization steps were

conducted at 4 °C (to avoid receptor degradation). Specifically 20 ml per t175-cm$^2$ flask of wash buffer (10 mM HEPES and 10 mM EDTA, pH 7.4) was added to the pellet. This was homogenized using an electrical homogenizer Ultra-Turrax (Ika-Werk GmbH & Co. KG, Staufen, Germany) (position 6, 4 × 5-s bursts) and subsequently centrifuged at 48,000×$g$ at 4 °C (Beckman Avanti J-251 Ultracentrifuge; Beckman Coulter, Fullerton, CA) for 30 min. The supernatant was discarded, and the pellet was re-homogenized and centrifuged as described above in wash buffer. The final pellet was suspended in ice-cold 10 mM HEPES and 0.1 mM EDTA, pH 7.4, at a concentration of 5–10 mg ml$^{-1}$. Protein concentration was determined using the bicinchoninic acid assay kit (Sigma-Aldrich), using BSA as a standard and aliquots maintained at −80 °C until required. Prior to their use, the frozen membranes were thawed and the membranes suspended in the assay buffer at a membranes concentration of 0.2 mg ml$^{-1}$.

**[$^3$H]-Spiperone saturation binding assays.** Increasing concentrations of [$^3$H]-spiperone (0.020–1.2 nM) were incubated with human $D_{2L}$ CHO cell membranes (10 µg per well) at 37 °C in assay binding buffer (20 mM HEPES 138 mM NaCl, 6 mM MgCl$_2$, 1 mM EGTA, and 1 mM EDTA pH 7.4) containing 100 µM GppNHp and 0.1% ascorbic acid in a 1 ml reaction volume. Non-specific binding was determined in the presence of 3 µM (+)-butaclamol. After a 2 h incubation period, bound and free [$^3$H]-spiperone were separated by fast-flow filtration through GF/B filters using a Filter Mate Harvester (PerkinElmer) followed by 2 ml wash with ice-cold PBS (Lonza). After drying, filter bound radioactivity was measured following addition of 40 µl of Microscint 20 (PerkinElmer) using a Topcount microplate scintillation counter (PerkinElmer). Aliquots of [$^3$H]-spiperone were also quantified accurately to determine how much radioactivity was added to each well using liquid scintillation spectrometry on a Tri-Carb liquid scintillation counter (PerkinElmer).

**Fluorescent ligand-binding assays.** All fluorescent binding experiments using PPHT-red were conducted in white 384-well Optiplate plates, in assay binding buffer, 20 mM HEPES, 138 mM NaCl, 6 mM MgCl$_2$, 1 mM EGTA, and 1 mM EDTA and 0.02% pluronic acid pH 7.4, 100 µM GppNHp, and 0.1% ascorbic acid. GppNHp was included to remove the G protein-coupled population of receptors that can result in two distinct populations of binding sites in membrane preparations, since the Motulsky–Mahan model[26] is only appropriate for ligands competing at a single site. In all cases, nonspecific binding was determined in the presence of 10 µM haloperidol.

**Determination of PPHT-red binding kinetics.** To accurately determine association rate ($k_{on}$) and dissociation rate ($k_{off}$) values, the observed rate of association ($k_{ob}$) was calculated using at least four different concentrations of PPHT-red. The appropriate concentration of PPHT-red was incubated with human $D_{2L}$ CHO cell membranes (2 µg per well) in assay binding buffer (final assay volume, 40 µl). The degree of PPHT-red bound to the receptor was assessed at multiple time points by HTRF detection to allow construction of association kinetic curves. The resulting data were globally fitted to the association kinetic model (Eq. 2) to derive a single best-fit estimate for $k_{on}$ and $k_{off}$ as described under data analysis.

**Competition binding kinetics.** To determine the association and dissociation rates of $D_2R$ ligands, we used a competition kinetic binding assay we recently described to profile the kinetics of a series of $D_2R$ agonists[21]. This approach involves the simultaneous addition of both fluorescent ligand and competitor to the receptor preparation, so that at $t = 0$ all receptors are unoccupied. 12.5 nM PPHT-red (a concentration which avoids ligand depletion in this assay volume, see Carter et al.,[47] was added simultaneously with the unlabeled compound (at $t = 0$) to CHO cell membranes containing the human $D_{2L}R$ (2 µg per well) in 40 µl of assay buffer. The degree of PPHT-red bound to the receptor was assessed at multiple time points by HTRF detection.

Nonspecific binding was determined as the amount of HTRF signal detected in the presence of haloperidol (10 µM) and was subtracted from each time point, meaning that $t = 0$ was always equal to zero. Each time point was conducted on the same 384-well plate incubated at 37 °C with orbital mixing (1 s of 100 RPM per cycle).

Multiple concentrations of unlabeled competitor were tested for determination of rate parameters. Data were globally fitted using Eq. (3) to simultaneously calculate $k_{on}$ and $k_{off}$. Different ligand concentration ranges were chosen, as compounds with a long residence time equilibrate more slowly, so a higher relative concentration is required to ensure the experiments reach equilibrium within a reasonable time frame (120 min), while still maintaining a good signal to noise.

**Signal detection and data analysis.** Signal detection was performed on a Pherastar FS (BMG Labtech, Offenburg, Germany) using standard HTRF settings. The terbium donor was always excited with three laser flashes at a wavelength of 337 nm. A kinetic TR-FRET signal was collected at 20 s intervals both at 665 and 620 nm, when using red acceptor. HTRF ratios were obtained by dividing the acceptor signal (665 nm) by the donor signal (620 nm) and multiplying this value by 10,000. Probe dissociation rates were analyzed by displacement of the tracer

with a large excess of an unlabeled ligand known to bind to the same site with similar or higher affinity.

All experiments were analyzed by non-regression using Prism 6.0 (GraphPad Software, San Diego, USA). Competition displacement binding data were fitted to sigmoidal (variable slope) curves using a "four parameter logistic equation":

$$Y = \text{Bottom} + (\text{Top} - \text{Bottom})/\left(1 + 10^{(\log \text{EC}_{50} - X)\text{Hillcoefficient}}\right). \quad (1)$$

IC$_{50}$ values obtained from the inhibition curves were converted to $K_i$ values using the method of Cheng and Prusoff[48]. PPHT-red association data were fitted as follows to a global fitting model using GraphPad Prism 6.0 to simultaneously calculate $k_{on}$ and $k_{off}$ using the following equation, where $k_{ob}$ equals the observed rate of association:

$$k_{ob} = [\text{PPHT} - \text{red}] \cdot k_{on} + k_{off}. \quad (2)$$

Association and dissociation rates for unlabeled antagonists were calculated using the equations described by Motulsky and Mahan[26]:

$$K_A = k_1[L] + k_2$$

$$K_B = k_3[I] + k_4$$

$$S = \sqrt{\left((K_A - K_B)^{2} + 4 \cdot k_1 \cdot k_3 \cdot L \cdot I \cdot 10^{-18}\right)}$$

$$K_F = 0.5 \cdot (K_A + K_B + S)$$

$$K_S = 0.5 \cdot (K_A + K_B - S)$$

$$\text{DIFF} = K_F - K_S \quad (3)$$

$$Q = \frac{B_{max} \cdot K_1 \cdot L \cdot 10^{-9}}{\text{DIFF}}$$

$$Y = Q \cdot \left(\frac{k_4 \cdot \text{DIFF}}{K_F \cdot K_S} + \frac{k_4 - K_F}{K_F} \cdot \exp^{(-K_F \cdot X)} - \frac{k_4 - K_S}{K_S} \cdot \exp^{(-K_S \cdot X)}\right)$$

Where: $X$ = Time (min), $Y$ = Specific binding (HTRF ratio 665 nm×620 nm×10,000), $k_1 = k_{on}$ PPHT-red, $k_2 = k_{off}$ PPHT-red, $L$ = Concentration of PPHT-red used (nM), $B_{max}$ = Total binding (HTRF ratio 665 nm/620 nm×10,000), $I$ = Concentration of unlabeled antagonist (nM). Fixing the above parameters allowed the following to be calculated: $k_3$ = Association rate of unlabeled ligand (M$^{-1}$ min$^{-1}$), $k_4$ = Dissociation rate of unlabeled ligand (min$^{-1}$). Dissociation of PPHT-red was fitted to a one phase mono-exponential decay function to estimate the dissociation rate of PPHT-red directly. Specific binding was determined by subtracting the nonspecific HTRF ratio from the total HTRF ratio.

**Modeling rebinding.** Rebinding describes the ability of a drug molecule to bind to multiple receptors within a compartment before diffusing away into bulk, the overall effect being extended target–receptor occupancy[24]. To examine this, we utilized a model of an immunological synapse with a compartment volume of 0.176 µm$^3$, which is within the range described for the dopamine synapse[49]. In this model, the overall macroscopic reversal rate ($k_r$) is described by the following equation:

$$k_r = k_{off}/(1 + k_{on} \cdot R/k_-), \quad (4)$$

where $k_{off}$ = dissociation rate from the receptor, $k_{on}$ = association rate onto the receptor, R = surface receptor density fixed at $1 \times 10^{11}$ cm$^{-2}$, and $k_-$ = the diffusion rate out of the synaptic compartment into bulk aqueous, fixed at $1.2 \times 10^{-5}$ cm s$^{-1}$. All data were analyzed using GraphPad Prism 6.0.

**Comparing binding kinetics and clinical side effect profile.** To explore the role of kinetics in determining on-target side effect liability, we correlated the kinetic values determined in this study with published clinical data taken from a comprehensive meta-analysis of clinically used APDs performed by Leucht and colleagues[31]. The majority of studies covered in the Leucht et al. meta-analysis included as subjects chronically ill patients with a history of APD exposure and the potential for modified $D_2R$ availability. To address this, we repeated the above analysis with reported data from available multiple-treatments studies in early psychosis patients who entered studies with minimal exposure to APDs[50–57]. Odds ratios for APD-induced EPS relative to spontaneous EPS in untreated patients were calculated as follows: % APD-treated with EPS × % untreated without EPS/% APD-treated without EPS × % untreated with spontaneous EPS. Unless otherwise stated, correlation analyses were performed using a two-tailed Spearman rank correlation allowing the calculation of the correlation coefficient, $r_s$. Although this analysis does not assume a linear relationship, a simple trend line has been

added to illustrate the positive or negative association between the two variables. Differences were considered significant at $P < 0.05$. All analysis were performed using GraphPad Prism 6.0.

**Data availability**. The data that support the findings of this study are available from the corresponding author on reasonable request.

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

## Acknowledgements

Thomas Roux (Cisbio Bioassays, Bagnols-sur-Cèze) for providing PPHT-red for the kinetic binding studies. Catherine Wark and Neil Cox (BMG Labtech Ltd.) for their expert technical assistance and BMG for personal sponsorship of David Sykes. H.M. is supported by a NIH Award P50 MH086404, New York State Office of Mental Health. J.A.J. is supported by Grants R01 MH54137 and K05 DA022413. The authors like to thank Dr. Scott Stroup for helpful discussion.

## Author contributions

D.A.S. conceived the project, developed the HTRF-based binding assay, analyzed data, established the utility of the rebinding model, and wrote the manuscript. H.M. sourced and compiled early psychosis patient findings, analyzed data, and wrote the manuscript. L.S. performed radioligand binding assays and analyzed data. N.H. production of the SNAP-tagged $D_2LR$ CHO cell line. J.A.J. helped coordinate the project and wrote the manuscript. J.R.L. helped coordinate the project and wrote the manuscript. S.J.C. conceived, coordinated, and supervised the project, provided guidance with analytical methods and wrote the manuscript.

## Additional information

**Competing interests:** The authors declare no competing financial interests.

