## [Peer Review File · Nature Communications]

Reviewers' comments:

Reviewer #1 (Remarks to the Author):

In this very interesting paper the authors have developed a novel assay, TR-FRET, to measure both the association and the dissociation rates of most available antipsychotics at the D2 dopamine receptor. In contrast to the fast-off hypothesis - one of the prevailing hypotheses which posits that rapid dissociation from the D2R accounts for reduced EPS associated with atypical antipsychotics - these authors have made the novel observation that the association rate rather than the dissociation rate is strongly correlated with EPS using the new assay. The enhanced association in the first episode population is particularly convincing. Prolactin elevation, on the other hand, correlated well with dissociation. These represent important new observations regarding the mechanism of antipsychotics and they should help improve in the search for new compounds with a profile that maximize efficacy while limiting D2 antagonist related side effects. The following relatively minor issues should be addressed:

- 1) In Figure 5 and Table 1, assigning molindone, loxapine, melperone and sulpiride to the atypical antipsychotic group is not a generally accepted convention even if it has been speculated that these may have "atypical" features. For example, molindone was used as a representative first generation antipsychotic in a well-conducted study which compared molindone, risperidone and olanzapine in about 120 youths with psychosis (Sikich et al, Am J Psychiatry, 2008). Molindone had significantly more EPS than the others. This doesn't mean that, with careful dosing, you can't limit EPS with molindone and loxapine, but that is true for other first generation antipsychotics as well. The available clinical evidence and clinical experience would suggest these 4 agents fall in the typical antipsychotic group.
- 2) The legend for Figure 3 appears to misstate what is indicated on the axes in panels A and B: $k(\text{off})$ and $k(\text{on})$ are interchanged in the text.
- 3) The Discussion should add a statement of limitation regarding the accuracy of the EPS data obtained from the Leucht meta-analysis. The degree of EPS associated with antipsychotics reported in clinical trials is almost always a secondary outcome measure and therefore usually less rigorously assessed and lacks checks on quality assurance (no specific training or certification for raters). In some studies, the use of anticholinergic medication is used as a proxy for EPS, making the assessment more indirect and dependent upon how carefully subjects were assessed for EPS and how aggressively it was treated. All these details are not needed but a general caveat about EPS data would be helpful to the reader, especially non-clinicians.

Reviewer #2 (Remarks to the Author):

The authors provide an intriguing counter-proposal to the hypothesis that the reduced EPS liability of some antipsychotic agents is can be attributed to their rapid dissociation from the D2 dopamine receptor (K_{off}). The authors determined the kinetics properties of a range of first (FGA) and second generation agents (SGA) using FRET to measure displacement of a fluorescent agonist from tagged, recombinantly expressed D2 receptors. A unique observation from these experiments was the wide range of K_{on} values found which contrasts to the prevailing view that on-rate is largely diffusion limited and therefore similar among antipsychotic agents and therefore, differences in receptor affinity among D2 antagonists is driven primarily by off-rates. In fact, K_{on} was most similar among FGA suggesting that differences in the affinity of these older agents may be more strongly influenced by K_{off} .

The authors next correlated their kinetic results with literature reports for two "on-target" side-effects of D2 receptor antagonists: extrapyramidal side-effect (EPS) and the elevation of plasma prolactin (PRL). In contrast to the "rapid dissociation hypothesis", the risk of EPS was correlated with K_{on} rather than K_{off} while PRL showed the opposite relationship. Thus authors contend that the risk of hyperprolactinemia is predicted by slow off-rates while EPS liability is predicted by fast on-rates. QT prolongation, an off-target side-effect (non-D2R), did not correlate with either kinetic

parameter.

The authors suggest that the spatial constraints of the synapse dictate that only antipsychotic agents with rapid on-rates can generate sufficient multiple drug-receptor interactions (rebinding) in the synapse to reduce dopaminergic signaling enough to produce EPS. Modeling both free versus limited diffusion conditions illustrated the dominant effect of K_{on} on occupancy was exaggerated in the limited diffusion condition. K_r , a reversal rate parameter calculated to reflect the local duration of antagonist activity within the synapse also correlated with EPS liability consistent with the suggestion that rapid rebinding is associated with sufficient receptor blockade to produce EPS. As predicted by the lack of correlation between K_{on} and PRL, K_r values failed to correlate with PRL.

This is very nice, very publishable study using new methodology to probe receptor/ligand interactions. The results are very relevant as they provide a satisfying alternative to a molecular model of EPS liability that has driven several drug discover efforts.

Questions:

1. Although the rebinding concept provides a feasible explanation for the dominant influence of K_{on} on the cumulative extent of D2 blockade and thereby EPS liability, it is difficult to see the value of K_r . Figures 3a and 4a show similar relationship between K_{on} and K_r to EPS liability and the generation of K_r for each agent did not improve the correlation to EPS. This was true for both the chronic (Leucht study) and first episode patient data (and includes the higher r value for the latter). The formula used to derive K_r does allow for any influence of K_{off} to be captured but this is apparently extremely minor. Perhaps the utility of K_r could be specifically addressed?
2. Given the emphasis on the spatial constraint for ligand diffusion provided by the pre and postsynaptic membranes, should some consideration be given to the former especially since the effects of presynaptic D2 blockade on synaptic levels of DA would oppose those of postsynaptic blockade?

Reviewer #3 (Remarks to the Author):

Review Manuscript: NCOMMS-16-24245

"Extrapyramidal side effects of antipsychotics are linked to their association kinetics at dopamine D2 receptors"

The manuscript describes a study on the binding kinetics of antipsychotic drugs on the D2 dopamine receptor (long form). Experimentally the study is relying on a recently developed FRET-based method to measure Agonist binding to D2 dopamine receptor heterologously expressed in CHO cells. Competition of PPHT-red (dopamine receptor agonist) by virtually all clinically used antipsychotics was used to calculate their k_{off} and k_{on} values. Based on the kinetic data, K_d -values were calculated and compared to K_i values determined under steady state conditions. Using these data, the authors analyse whether or not K_{on} or K_{off} values of the different antipsychotic drugs correlate with clinically relevant D2 dependent side effects, specifically EPS syndrome and prolactin levels. Clinical data were taken from a large meta analysis from Leucht et al. 2013 (Lancet). In addition the authors analysed a smaller set of clinical studies including only patients that received their first antipsychotic therapy. The novelty of the present study are:

-Analysis of binding kinetics of antipsychotic drugs in a comparable setting. Finding of no significant correlation of the dissociation kinetics with EPS side effects. In contrast, a significant inverse correlation of the k_{off} of these drugs with the prolactinomic side effects were found. Even more, the authors found a positive correlation of EPS side effects with the k_{on} of antipsychotic drug binding to D2 receptors.

These experimental results were interpreted by theoretical considerations that took into account the differential influence of k_{on} and k_{off} on rebinding in a diffusion restricted compartment, which is much more likely to play a major role in the synaptic cleft compared to the pituitary which connects to the portal vein system.

Comments:

Major comments:

1.) The study presents solid data on the binding kinetics of many of the clinically used antipsychotics to D2 dopamine receptors, showing that not only dissociation kinetics vary a lot between different drugs but also association kinetics. The clinically relevant D2R-mediated side effects correlate differentially with association kinetics and dissociation kinetics. Based on the data shown I would argue that the data clearly show that there is no clear correlation between incidence of EPS and koff, and also that there is no clear correlation between kon and prolactin levels. However, the correlation that was found for Kon and EPS incidence and the one of Koff and Prolactin increase seem to be less obvious, even though statistically significant. For instance in Fig. 3a, there are many drugs with indistinguishable EPS odds ratio, however more than 100 fold differences in the kon value.
2. Why did the authors left out Lurasidone, which is included in the meta analysis the study refers to. Since it exhibits quite some effects in both systems (EPS incidence and Prolactin increases) it would be of quite some interest to include. It could strengthen the claim of the authors.
3. If the affinity of the antipsychotic drug to D2R correlates very well to the antipsychotic effect. The authors make a valid claim that in the case of the pituitary rebinding events will be of much less relevance compared to synapsis such as those controlling EPS. The dopaminergic synapsis that mediate the antipsychotic activity of antipsychotics should have the same size as those of controlling EPS. Shouldn't we see a perfect correlation of Koff with antipsychotic activity?
4. I'm not so sure whether it is legitimate to discuss Aripiprazole as an "outlier" because of its very weak partial agonistic profile. At the end, it does act as an antagonist of dopamine.
5. The authors solely relay on competition of an antipsychotic drug from PPHT-red activated receptors. Based on structural and dynamic knowledge it seems possible that kon values might be different depending whether a receptor was activated by agonist compared to empty receptors. It would be nice if at least for a very few cases the authors could compare binding kinetics of a drug in the absence and presence of an agonist.

Specific comments: "Aripiprazole" seems to be the right name instead of "Apripiprazole"
(Figure 3)

Reviewers' comments:

Reviewer #1 (Remarks to the Author):

In this very interesting paper the authors have developed a novel assay, TR-FRET, to measure both the association and the dissociation rates of most available antipsychotics at the D2 dopamine receptor. In contrast to the fast-off hypothesis - one of the prevailing hypotheses which posits that rapid dissociation from the D2R accounts for reduced EPS associated with atypical antipsychotics – these authors have made the novel observation that the association rate rather than the dissociation rate is strongly correlated with EPS using the new assay. The enhanced association in the first episode population is particularly convincing. Prolactin elevation, on the other hand, correlated well with dissociation. These represent important new observations regarding the mechanism of antipsychotics and they should help improve in the search for new compounds with a profile that maximize efficacy while limiting D2 antagonist related side effects. The following relatively minor issues should be addressed:

1) In Figure 5 and Table 1, assigning molindone, loxapine, melperone and sulpiride to the atypical antipsychotic group is not a generally accepted convention even if it has been speculated that these may have "atypical" features. For example, molindone was used as a representative first generation antipsychotic in a well-conducted study which compared molindone, risperidone and olanzapine in about 120 youths with psychosis (Sikich et al, Am J Psychiatry, 2008). Molindone had significantly more EPS than the others. This doesn't mean that, with careful dosing, you can't limit EPS with molindone and loxapine, but that is true for other first generation antipsychotics as well. The available clinical evidence and clinical experience would suggest these 4 agents fall in the typical antipsychotic group.

We thank the reviewer for this supportive review and for this useful comment. The current 'typical' versus 'atypical' classification, rather than having an underlying mechanistic basis, is based upon clinical observations that are inherently subject to additional factors such as dosing regimen that may be inconsistent across different studies. Thus, as the reviewer aptly points out, drugs that have been classed by one study as typical are described as atypical in other studies. The use of a model based on the best pharmacological predictors for a given side effect would represent a rational approach to optimizing APD tolerability so that relative risk for EPS and hyperprolactemia can be independently estimated across drugs. To address this need, in this study we provide evidence that ligand binding kinetics and more specifically rebinding rate can predict the propensity of D₂R targeted antipsychotic drugs to cause EPS and hyperprolactemia, and provide evidence that APD rebinding within the synapse may provide the underlying mechanism for this observation. We believe, however, we must acknowledge the historic classifications of the drugs we examined. Taking the reviewers' point on molindone we have opted to add a 3rd group (typical/atypical), wherein we include drugs that have been classified as both "atypical" and typical in different studies. They include sulpiride, thioridazine, molindone, loxapine, raclopride, melperone, and zotepine. Thus this grouping more accurately reflects historical observations and we believe that regrouping and incorporating additional references suggested by the reviewer brings more balance and accuracy to our discussion of these drugs.

2) The legend for Figure 3 appears to misstate what is indicated on the axes in panels A and B: k(off) and k(on) are interchanged in the text.

We thank the reviewer for noticing this mistake. We have now rectified it.

3) The Discussion should add a statement of limitation regarding the accuracy of the EPS data obtained from the Leucht meta-analysis. The degree of EPS associated with antipsychotics reported in clinical trials is almost always a secondary outcome measure and therefore usually less rigorously assessed and lacks checks on quality assurance (no specific training or certification for raters). In some studies, the use of anticholinergic medication is used as a proxy for EPS, making the assessment more indirect and dependent upon how carefully subjects were assessed for EPS and how aggressively it was treated. All these details are not needed but a general caveat about EPS data would be helpful to the reader, especially non-clinicians.

We agree with the reviewer and thank them for highlighting this issue. As suggested we have added a comment in the discussion to acknowledge this caveat:

While meta-analytic methods assure the best control for the quality of the data across studies, it is important to note that because the primary focus of these studies was efficacy, EPS was often not a well-controlled outcome. Indeed, the validation of the present model, like preceding hypotheses on EPS, is limited by the relative paucity of studies comparing antipsychotic drugs in which EPS is a primary outcome that is controlled for the minimum dose required to treat the psychosis and for adjunct treatments. To address this we performed an additional analysis of studies in which EPS was systematically measured in drug free, patients with early psychosis in which adjunct therapy was minimal and using data from groups within a narrow range of doses that were the closest to the reported minimum effective dose for each drug.

Reviewer #2 (Remarks to the Author):

The authors provide an intriguing counter-proposal to the hypothesis that the reduced EPS liability of some antipsychotic agents is can be attributed to their rapid dissociation from the D2 dopamine receptor (Koff). The authors determined the kinetics properties of a range of first (FGA) and second generation agents (SGA) using FRET to measure displacement of a fluorescent agonist from tagged, recombinantly expressed D2 receptors. A unique observation from these experiments was the wide range of K_{on} values found which contrasts to the prevailing view that on-rate is largely diffusion limited and therefore similar among antipsychotic agents and therefore, differences in receptor affinity among D2 antagonists is driven primarily by off-rates. In fact, K_{on} was most similar among FGA suggesting that differences in the affinity of these older agents may be more strongly influenced by Koff. The authors next correlated their kinetic results with literature reports for two "on-target" side-effects of D2 receptor antagonists: extrapyramidal side-effect (EPS) and the elevation of plasma prolactin (PRL). In contrast to the "rapid dissociation hypothesis", the risk of EPS was correlated with K_{on} rather than Koff while PRL showed the opposite relationship. Thus authors contend that the risk of hyperprolactinemia is predicted by slow off-rates while EPS liability is predicted by fast on-rates. QT prolongation, an off-target side-effect (non-D2R), did not correlate with either kinetic parameter.

The authors suggest that the spatial constraints of the synapse dictate that only antipsychotic agents with rapid on-rates can generate sufficient multiple drug-receptor interactions (rebinding) in the synapse to reduce dopaminergic signaling enough to produce EPS. Modeling both free versus limited diffusion conditions illustrated the dominant effect of K_{on} on occupancy was exaggerated in the limited diffusion condition. K_r , a reversal rate parameter calculated to reflect the local duration of antagonist activity within the synapse also correlated with EPS liability consistent with the suggestion that rapid rebinding is associated with sufficient receptor blockade to produce EPS. As predicted by the lack of correlation between K_{on} and PRL, K_r values failed to correlate with PRL.

This is very nice, very publishable study using new methodology to probe receptor/ligand interactions. The results are very relevant as they provide a satisfying alternative to a molecular model of EPS liability that has driven several drug discover efforts.

Questions:

1. Although the rebinding concept provides a feasible explanation for the dominant influence of K_{on} on the cumulative extent of D2 blockade and thereby EPS liability, it is difficult to see the value of K_r . Figures 3a and 4a show similar relationship between K_{on} and K_r to EPS liability and the generation of K_r for each agent did not improve the correlation to EPS. This was true for both the chronic (Leucht study) and first episode patient data (and includes the higher r value for the latter). The formula used to derive K_r does allow for any influence of Koff to be captured but this is apparently extremely minor. Perhaps the utility of K_r could be specifically addressed?

We thank the reviewer for their supportive comments. We agree that we did not make the utility of the parameter K_r sufficiently clear in the previous version of this manuscript and have addressed this in this revised version. Although the correlation between k_{on} and EPS is strong, this correlation does not provide any insight into the mechanism that might underlie this observation. One mechanism whereby k_{on} might influence D2R receptor occupancy is through rebinding. This is likely to be particularly relevant within the small compartment of a synapse in which drug diffusion is limited and thus the local concentration of drug is likely to be high. To explore this possibility we built a model described by the equation:

$$K_r = k_{off} / (1 + k_{on} * [R] / k_-)$$

where: k_r = the resulting overall reversal rate for blockade of synaptic receptors, k_{off} = dissociation rate from the receptor, k_{on} = association rate onto the receptor, $[R]$ = surface receptor density

and k_{-} = the diffusion rate out of the synaptic compartment into bulk aqueous. The reviewer is correct that although the model also includes k_{off} , it appears that, for our chosen set of conditions and compounds, k_{off} plays a relatively minor role in the overall reversal rate. Thus, in our synapse model there is no real improvement in correlation with k_r over k_{on} . In other systems this might not be the case. For example, changes in the dimensions or receptor density of the compartment might reduce the impact of rebinding, meaning k_r would be largely driven by k_{off} . In these situations, it would be expected that k_r would be better correlated to EPS than k_{on} . Thus, if one knows the kinetic parameters of a certain drug, one can use k_r to predict the receptor occupancy of this drug under different degrees of limited diffusion. We have clarified this point in the text.

2. Given the emphasis on the spatial constraint for ligand diffusion provided by the pre and postsynaptic membranes, should some consideration be given to the former especially since the effects of presynaptic D2 blockade on synaptic levels of DA would oppose those of postsynaptic blockade?

We enthusiastically agree that modelling pre and post-synaptic components separately is a natural next step in the evolution of this model. In this study, however, we treat the whole synapse as a compartment so cannot distinguish between pre- and post-synaptic receptors. Once drug is released from receptors on one membrane it is free to bind either pre- or post-synaptically, so it would be expected that both receptor populations would be occupied by drug to a similar degree. As a consequence, although pre-synaptic rebinding may result in higher DA concentrations in the synapse, the post-synaptic receptors will also be occupied, likely blocking the action of released DA.

Reviewer #3 (Remarks to the Author):

Review Manuscript: NCOMMS-16-24245: "Extrapyramidal side effects of antipsychotics are linked to their association kinetics at dopamine D2 receptors"

The manuscript describes a study on the binding kinetics of antipsychotic drugs on the D2 dopamine receptor (long form). Experimentally the study is relying on a recently developed FRET-based method to measure Agonist binding to D2 dopamine receptor heterologously expressed in CHO cells. Competition of PPHT-red (dopamine receptor agonist) by virtually all clinically used antipsychotics was used to calculate their koff and kon values. Based on the kinetic data, Kd-values were calculated and compared to Ki values determined under steady state conditions. Using these data, the authors analyse whether or not Kon or Koff values of the different antipsychotic drugs correlate with clinically relevant D2 dependent side effects, specifically EPS syndrome and prolactin levels. Clinical data were taken from a large meta analysis from Leucht et al. 2013 (Lancet). In addition the authors analysed a smaller set of clinical studies including only patients that received their first antipsychotic therapy. The novelty of the present study are:

- Analysis of binding kinetics of antipsychotic drugs in a comparable setting. Finding of no significant correlation of the dissociation kinetics with EPS side effects. In contrast, a significant inverse correlation of the koff of these drugs with the prolactinomic side effects were found. Even more, the authors found a positive correlation of EPS side effects with the kon of antipsychotic drug binding to D2 receptors.

These experimental results were interpreted by theoretical considerations that took into account the differential influence of kon and koff on rebinding in a diffusion restricted compartment, which is much more likely to play a major role in the synaptic cleft compared to the pituitary which connects to the portal vein system.

Comments:

Major comments:

1.) The study presents solid data on the binding kinetics of many of the clinically used antipsychotics to D2 dopamine receptors, showing that not only dissociation kinetics vary a lot

between different drugs but also association kinetics. The clinically relevant D2R-mediated side effects correlate differentially with association kinetics and dissociation kinetics. Based on the data shown I would argue that the data clearly show that there is no clear correlation between incidence of EPS and k_{off} , and also that there is no clear correlation between k_{on} and prolactin levels. However, the correlation that was found for k_{on} and EPS incidence and the one of k_{off} and Prolactin increase seem to be less obvious, even though statistically significant. For instance in Fig. 3a, there are many drugs with indistinguishable EPS odds ratio, however more than 100 fold differences in the k_{on} value.

We acknowledge the reviewer's concern. However, in our view, statistical analysis is the only independent test we have of a relationship between two parameters and we have used this to guide our conclusions. The correlation of EPS with k_{on} is statistically significant. However, as highlighted by reviewer 1, such a correlation will likely be influenced by the inherent variability in the clinical data. In addition many of these compounds also display activity at other receptors (eg 5HT2A antagonism) that may make an additional contribution to their lower propensity to cause EPS and thus could cause deviation from the rank order discussed here. We have been careful to acknowledge these caveats in the text. Indeed, given these confounding factors, we feel that the significant correlation between EPS and k_{on} or k_r is a noteworthy finding. For clarity we have now specifically stated in the figure legends that a two-tailed Spearman rank correlation was performed (which does not assume linearity) and that the lines depicted demonstrate a trend. We feel the correlation of k_{off} with prolactin secretion is strong considering we compare clinical data obtained in an open system with in-vitro binding data from a closed system. As discussed above, future studies in which EPS is a primary outcome, that is controlled both for the minimum dose required to treat psychosis and for adjunct treatments, will provide an important resource with which to validate our model further.

2. Why did the authors left out Lurasidone, which is included in the meta analysis the study refers to. Since it exhibits quite some effects in both systems (EPS incidence and Prolactin increases) it would be of quite some interest to include. It could strengthen the claim of the authors.

Lurasidone would indeed be suitable for inclusion in this study. However, this compound was not included as we were unable to obtain it when the experiments were conducted.

3. If the affinity of the antipsychotic drug to D2R correlates very well to the antipsychotic effect. The authors make a valid claim that in the case of the pituitary rebinding events will be of much less relevance compared to synapsis such as those controlling EPS. The dopaminergic synapsis that mediate the antipsychotic activity of antipsychotics should have the same size as those of controlling EPS. Shouldn't we see a perfect correlation of k_{off} with antipsychotic activity?

The reviewer makes an interesting point. However, because these drugs were dosed in trials to achieve antipsychotic efficacy, this measurement becomes the baseline. Indeed, in this meta-analysis there was very little difference in efficacy observed across the cohort, making it difficult to draw conclusions on comparative antipsychotic activity. For this reason we have avoided commenting on the correlation between k_{on} or k_{off} and efficacy.

4. I'm not so sure whether it is legitimate to discuss Aripiprazole as an "outlier" because of its very weak partial agonistic profile. At the end, it does act as an antagonist of dopamine.

We agree with the reviewer that, in conditions of high dopaminergic tone a weak partial agonist such as aripiprazole would antagonize synaptic dopamine signalling. However, the degree of this blockade will be partial to a level determined by the intrinsic efficacy of the partial agonist, and the receptor reserve and signal coupling efficiency of the system. Aripiprazole is the only APD included in this study that has been shown to display partial agonism at the D2R (Shapiro et al., 2003; Davies et al., 2004 and Burns 2002). Indeed, many researchers have suggested aripiprazole belongs in a third class of D2R targeted APDs (Stahl 2001). While, to our knowledge, there has been no clear demonstration of a relationship between this partial agonism and its low propensity to cause EPS, it is interesting to note that there is clinical evidence that aripiprazole decreases prolactin release, unlike all other APDs. Therefore, because of this evidence for a different mechanism of action at the D2R we excluded it from the correlation between the kinetic parameters of pure D2 receptor antagonists and their propensity to cause EPS and prolactin increase.

5. The authors solely rely on competition of an antipsychotic drug from PPHT-red activated receptors. Based on structural and dynamic knowledge it seems possible that k_{on} values might be different depending whether a receptor was activated by agonist compared to empty receptors. It would be nice if at least for a very few cases the authors could compare binding kinetics of a drug in the absence and presence of an agonist.

We agree with the reviewer that the dissociation rate of a drug could be quite different from an active (G protein-coupled receptor) as compared to an inactive (G protein-uncoupled receptor). In particular, agonists would be predicted to bind with high affinity to a G protein-coupled receptor but with lower affinity to a G protein-uncoupled receptor. To address this issue all experiments were conducted in the presence of GppNHp to uncouple G protein from the receptor, forcing a low affinity receptor conformation. Thus, the kinetic parameters measured for our agonist tracer ligand and unlabelled ligands of interest reflect those at an inactive G protein-uncoupled receptor only. However, to satisfy the reviewers concern we have also compared our original kinetics for a selection of compounds with those generated using a fluorescent analogue of the D2 antagonist clozapine (F-clozapine). As can be seen in the table below, the calculated off rates of these compounds are similar, suggesting there is little influence of the tracer on these kinetic parameters.

Table of k_{off} values determined with either F-clozapine or F-PPHT as the tracer molecule:

Compound	F-PPHT		F-Clozapine	
	k_{off} (min ⁻¹)	k_{on} (M ⁻¹ min ⁻¹)	k_{off} (min ⁻¹)	k_{on} (M ⁻¹ min ⁻¹)
Molindone	1.69	8.69×10^7	1.69	9.76×10^7
Paliperidone	0.44	1.88×10^8	0.45	2.18×10^8
Sulipride	2.23	1.60×10^8	1.18	9.23×10^7
Remoxipride	1.90	1.16×10^7	1.66	1.17×10^7
Melperone	1.48	1.99×10^7	1.10	8.76×10^8
Clozapine	1.67	8.23×10^7	1.84	4.60×10^7

Specific comments: "Aripiprazole" seems to be the right name instead of "Apripiprazole" (Figure 3)

Thank you for spotting this error in the figures; this has now been amended.

REVIEWERS' COMMENTS:

Reviewer #1 (Remarks to the Author):

The authors have satisfactorily addressed each of the points raised in the earlier review. This paper makes a novel and important contribution to our understanding of the mechanism of action of antipsychotic medications, especially in regard to D2-related side effects.

Remaining item: typo in Table 1, trifluoperazine is missing the "o".

Reviewer #3 (Remarks to the Author):

The authors resubmitted a revised version of their manuscript, which addresses important points raised by three reviewers. Importantly, they responded to all comments with detailed responses, which were satisfactory for at least the points raised by myself. I specifically appreciate the measurements of k_{on} and k_{off} also for competition with labeled clozapine (table shown to reviewer). The only data in that table that is puzzling to me is the k_{on} for melperone, which seems to be (approx. 40 fold) different dependent on whether F-Clozapine or F-PPHT was used. Is this real or a typo?

Overall the manuscript in its present form has much improved. I'd like to congratulate the authors on a very interesting paper that indicates how important the binding kinetics of drugs can be.